# RGANet: A Human Activity Recognition Model for Extracting Temporal and Spatial Features from WiFi Channel State Information

**DOI:** 10.3390/s25030918

**Published:** 2025-02-03

**Authors:** Jianyuan Hu, Fei Ge, Xinyu Cao, Zhimin Yang

**Affiliations:** School of Computer Science, Central China Normal University, Wuhan 430070, China; hujianyuan@mails.ccnu.edu.cn (J.H.); caoxinyu@mails.ccnu.edu (X.C.); zhiminyang@mails.ccnu.edu.cn (Z.Y.)

**Keywords:** Human Activity Recognition (HAR), Channel State Information (CSI), Deep Learning (DL)

## Abstract

With the rapid advancement of communication technologies, wireless networks have not only transformed people’s lifestyles but also spurred the development of numerous emerging applications and services. Against this backdrop, research on Wi-Fi-based human activity recognition (HAR) has become a hot topic in both academia and industry. Channel State Information (CSI) contains rich spatiotemporal information. However, existing deep learning methods for human activity recognition (HAR) typically focus on either temporal or spatial features. While some approaches do combine both types of features, they often emphasize temporal sequences and underutilize spatial information. In contrast, this paper proposes an enhanced approach by modifying residual networks (ResNet) instead of using simple CNN. This modification allows for effective spatial feature extraction while preserving temporal information. The extracted spatial features are then fed into a modifying GRU model for temporal sequence learning. Our model achieves an accuracy of 99.4% on the UT_HAR dataset and 99.24% on the NTU-FI HAR dataset. Compared to other existing models, RGANet shows improvements of 1.21% on the UT_HAR dataset and 0.38% on the NTU-FI HAR dataset.

## 1. Introduction

With the rise of concepts such as smart homes and smart cities, an increasing number of devices are being connected to the Internet, forming a vast Internet of Things (IoT). In this context, people have begun to explore how to use these interconnected devices to enhance the quality of life and safety levels. Wi-Fi, as one of the most commonly used wireless communication technologies, is ubiquitous in home and work environments, making it an ideal platform for Human Activity Recognition (HAR).

HAR is a technology that analyzes information about human movements obtained from various sensors using signal processing methods [1]. It is a practical technology widely applied in areas such as password input detection [2], user authentication [3], fall detection [4], human presence detection [5], human structure modeling [6], and smart furniture [7].

However, traditional HAR methods typically rely on wearable sensors like accelerometers and gyroscopes [8] or cameras [9]. These approaches have some drawbacks. For instance, using wearable sensors for recognition requires users to wear additional devices, and the analysis of the recorded information from these devices is necessary to identify actions [10]. Moreover, some of these devices can be expensive. Using high-resolution cameras for recognition involves recording the motion information of the detection target through photos or videos and then processing features with image processing, deep learning, and other techniques to complete action recognition [11]. Although this method offers higher recognition accuracy, it requires suitable lighting conditions and may lead to privacy issues such as the leakage of facial information.

HAR based on Wi-Fi sensing technology has attracted significant attention from researchers. Compared to traditional HAR, Wi-Fi sensing can perform action recognition without disturbing users, independent of lighting conditions, while preserving personal privacy and reducing deployment costs. In recent years, the development of new technologies such as Multiple-Input Multiple-Output (MIMO) [12], millimeter-wave radar, and Channel State Information (CSI) have enabled Wi-Fi signals to carry more information about environmental changes. This provides a possibility for more accurately capturing human motion characteristics. As technology advances and living standards improve, Wi-Fi has become an integral part of daily life, almost to the point of being a necessity.

Wi-Fi sensing is an emerging technology that leverages Wi-Fi signals to perceive and interpret the surrounding environment, offering a variety of applications such as gesture recognition [13], human modeling [14], occupancy detection [15], fall detection [16], human identification [17], and people counting [18]. Despite these advancements, the interpretation of Wi-Fi signals and the extraction of meaningful insights from them continue to pose a significant challenge. Our research on human activity recognition based on Wi-Fi is indeed one of the applications within the realm of Wi-Fi sensing. Existing studies on Wi-Fi-based human activity recognition primarily utilize two types of signals found in Wi-Fi: RSSI (Received Signal Strength Indicator) and CSI. RSSI is a crucial metric of signal strength in wireless communication systems and is widely utilized across various wireless technologies, including Wi-Fi, Bluetooth, Zigbee, and others. It enables devices to evaluate link quality, optimize communication parameters, and, in certain scenarios, can be used for positioning and tracking [19]. RSSI’s simplicity, low cost, and minimal power consumption [20] have made it a popular choice for many applications. However, as wireless communication technologies have advanced, the limitations of RSSI have become more apparent, especially in complex environments where factors such as multipath effects, interference, and environmental changes can significantly impact its reliability.

To achieve a more precise characterization of the wireless channel, CSI has been introduced. CSI provides a detailed snapshot of the wireless channel by capturing information about the amplitude and phase of the received signal at each subcarrier within an Orthogonal Frequency-Division Multiplexing (OFDM) system. This granular data allows for a deeper understanding of the channel conditions, enabling more precise activity recognition, improved performance in adaptive transmission techniques, and enhanced accuracy in localization and tracking applications [21].

As a result, in scenarios requiring high precision and robustness, such as human activity recognition, indoor positioning, and smart environment monitoring, CSI has demonstrated superior reliability and accuracy in these tasks compared to RSSI [22,23].

Machine Learning (ML) and Deep Learning (DL) are critical technologies for achieving human recognition using CSI. However, ML alone is not sufficient to handle replicated CSI information; it requires well-extracted features to be effective. Therefore, the current predominant research approach is based on DL, which can automatically learn features from raw data. Current DL studies primarily employ algorithms such as Convolutional Neural Networks (CNN), Gated Recurrent Unit (GRU), Long Short-Term Memory network (LSTM), and combinations like CNN+LSTM. CSI contains temporal features and spatial features. Temporal features reflect how CSI signals change over time due to human activities and are useful for detecting activities that have characteristic time patterns. Spatial features reflect how CSI signals differ across space (i.e., different antennas) and are useful for determining the location and movement patterns of individuals.

Most of these studies focus on extracting either spatial features or temporal features independently, with fewer approaches integrating both. The methods that combine spatial and temporal feature extraction often use a simple CNN for spatial features alongside various improved models for capturing temporal characteristics. Therefore, we propose the RGANet model, which employs a modified ResNet for spatial feature extraction and an enhanced GRU model for temporal feature extraction, to improve the model’s capability in recognizing human actions. The main contributions of this paper are summarized as follows:
We introduce a deep learning-based model (RGANet) that can simultaneously leverage both temporal and spatial features, achieving high-precision recognition of human actions.In the spatial feature extraction component of the RGANet model, we utilize depthwise separable convolutions to reduce the computational cost of the ResNet architecture while maintaining model accuracy. Furthermore, we introduce the scSE (spatial and channel Squeeze-and-Excitation) module to further enhance the spatial feature extraction capabilities of the model.For the temporal feature extraction part of the RGANet model, we employ a GRU model, which has lower computational costs compared to LSTM. We also augment it with a simplified attention mechanism to strengthen the model’s ability to capture temporal information.

## 2. Related Work

The application of CSI in HAR is a method that leverages the physical layer information of Wi-Fi signals to detect and classify human movements. This approach does not rely on traditional wearable sensors or cameras but instead infers human activities by analyzing the CSI data contained within the data packets transmitted between Wi-Fi devices. The process is illustrated in Figure 1 below.

### 2.1. Background of RSSI

RSSI is a metric used in wireless communication systems to indicate the strength of received radio signals. It is typically measured in decibel-milliwatts (dBm), representing the logarithmic ratio of the received signal power relative to 1 milliwatt. The closer the RSSI value is to 0 dBm, the stronger the signal; conversely, the more negative the value (the larger its absolute value), the weaker the signal. In everyday use, common wireless technologies such as Wi-Fi and Bluetooth utilize RSSI to determine signal strength. The received signal strength can be calculated using the following formula:(1)$$x\left(dBm\right)=10{log}_{10}(\frac{P(mW)}{1\;mW})$$
where $x\left(dBm\right)$ represents the signal strength in decibel-milliwatts (dBm) and P(mW) denotes the received signal power in milliwatts (mW). The reference power is typically set to 1 mW.

In wireless communication systems, RSSI is a crucial signal strength metric widely used in various wireless technologies, such as Wi-Fi, Bluetooth, Zigbee, and more. It assists devices in evaluating link quality and optimizing communication parameters, and in some scenarios, it is utilized for positioning and tracking. Despite its simplicity and the low cost and energy consumption associated with RSSI, which have led to its widespread application, the evolution of wireless communication technologies has begun to highlight its limitations, especially in complex environments. To more accurately capture the characteristics of wireless channels, CSI has emerged and is gradually replacing RSSI in certain application scenarios.

### 2.2. Background of CSI

CSI is a critical parameter in wireless communication systems used to describe the characteristics of the channel. Unlike RSSI, CSI provides richer channel information, including the amplitude and phase of each subcarrier, frequency domain characteristics, spatial correlation, and more [24]. Through CSI, the receiver can gain a more accurate understanding of the channel state, thereby optimizing signal processing, improving communication quality, and enabling advanced applications. Figure 2 shows the waveform of a single-antenna carrier after processing the CSI information.

From the perspective of application location, CSI can be categorized into transmitter CSI and receiver CSI. CSI encompasses a wealth of channel characteristic information, such as the effects of distance, scattering, fading, and other factors on the signal. In wireless communication systems, CSI is typically represented by the channel state matrix H, which is a collection of information for each subcarrier signal. Specifically, the mathematical model of CSI can be expressed as follows:(2)Y=HX+N

Here, Y is the received signal vector, X is the transmitted signal vector, N is the Additive White Gaussian Noise (AWGN), and H is the channel state matrix, which represents the CSI and contains the phase and amplitude information for each subcarrier [25]. The channel matrix H can be further expanded as follows:(3)$$H=\left[H_{1},H_{2},H_{3},\ldots H_{K}\right]$$

Here, K denotes the number of subcarriers, and each $H_{i}$ represents the phase and amplitude information for a single subcarrier. A key parameter for evaluating the quality of CSI data is the number of subcarriers, which is determined by the bandwidth and the tools used. Generally, the more subcarriers there are, the higher the resolution of channel estimation, allowing for more precise capture of multipath effects and other channel characteristics.

In Wi-Fi communication, CSI reflects the propagation characteristics of wireless signals after they undergo diffraction, reflection, and scattering in the physical environment, describing the channel properties of the communication link. For modern wireless communication networks that adhere to the IEEE 802.11 standard, MIMO and OFDM technologies at the physical layer help increase the data capacity and orthogonality of transmission channels affected by multipath propagation [26]. As a result, modern Wi-Fi access points are typically equipped with multiple antennas and use multiple OFDM subcarriers to enhance transmission performance.

For each pair of transmit and receive antennas, CSI describes the multipath effects, amplitude attenuation, and phase shift on each subcarrier. Compared to RSSI, CSI provides higher sensing resolution, enabling a finer capture of channel characteristics. Specifically, the Channel Impulse Response (CIR) of Wi-Fi signals is the superposition of all multipath components in the wireless channel and can be expressed as follows:(4)$$h\left(t\right)=\sum_{i=1}^{L}a_{i}e^{j\phi_{i}}\delta \left(t-t_{i}\right)$$

Here, $a_{i}$ denotes the amplitude of the ith multipath component, $\phi_{i}$ denotes the phase of the ith multipath component, titi denotes the time delay of the ith multipath component, ith represents the total number of paths, and j is the imaginary unit. $\delta \left(t-t_{i}\right)$ is the idealized impulse function, also known as the Dirac delta function, which indicates an instantaneous pulse at the moment $t=t_{i}$. The role of this function is to concentrate the multipath components at the specific time point $t_{i}$.

The Channel Frequency Response (CFR), $H\left(f\right)$, is typically obtained by performing a Fourier Transform on the CIR in the time domain, that is(5)$$H\left(f\right)=\int_{-\infty }^{\infty }h\left(t\right)e^{-j2\pi ft}d_{t}$$
where f is the frequency variable.

In MIMO and OFDM systems, CSI is typically represented as a complex matrix, where each element $h_{N_{T}N_{R}k}$ denotes the channel response from one transmit antenna to one receive antenna on a specific subcarrier. By decomposing each complex value into magnitude and phase, this polar representation is widely used in MIMO and OFDM systems for channel estimation, beamforming, adaptive modulation and coding, and other techniques, significantly enhancing system performance. The decomposition formula for $h_{N_{T}N_{R}k}$ is as follows:(6)$$h_{N_{T}N_{R}k}=\left|h_{N_{T}N_{R}k}\right|e^{j∠h_{N_{T}N_{R}k}}$$
where $h_{N_{T}N_{R}k}$ represents the channel response from the $N_{T}$th transmit antenna to the $N_{R}$th receive antenna on the kth subcarrier, $\left|h_{N_{T}N_{R}k}\right|$ is the magnitude of $h_{N_{T}N_{R}k}$, representing the amplitude of the channel response, and $∠h_{N_{T}N_{R}k}$ is the phase of $h_{N_{T}N_{R}k}$, representing the phase shift introduced by the channel.

### 2.3. WiFi CSI-Based HAR

In recent years, deep learning techniques have achieved remarkable success in areas such as computer vision and Natural Language Processing (NLP). This has also prompted many researchers to apply these techniques to WiFi-based HAR. Among them, CNNs, commonly used in the field of computer vision, were first applied to the processing of CSI signals. In 2018, Wen et al. [27] treated the CSI data as images for processing and used CNNs to encode and decode the CSI data. In addition, Wang et al. [28] proposed an indoor positioning and HAR system, creating a dataset for six different types of human activities. They introduced a multitask 1D CNN based on the ResNet architecture. Their proposed model achieved an accuracy rate of 95.68%. Yang et al. [29] developed a HAR framework using WiFi CSI signals. They first proposed an algorithm for automatically selecting antennas based on their sensitivity to various activities and evaluated their approach using three machine learning classifiers and a CNN model, achieving an average accuracy of 96.8%. Although CNNs have achieved some promising results in HAR tasks based on CSI, CNNs are typically designed to process two-dimensional image data. CSI data, however, are usually in the form of high-dimensional time series or matrices. Directly feeding CSI data into a standard CNN architecture may lead to suboptimal outcomes because CNNs might not effectively capture long-term dependencies within the time series. For data with long time spans, CNNs may fail to adequately preserve temporal information.

Subsequently, researchers have also applied time series models to HAR tasks based on CSI. Damodaran et al. [30] created a human activity dataset comprising only four classes and utilized LSTM and SVM for classification tasks. Their results showed that LSTM achieved higher accuracy than SVM. Ding et al. [31] collected a dataset that includes two environments and six activities. They used an RNN architecture integrated with an LSTM module for recognition tasks and achieved an accuracy of over 95% for each class of activity. In their work, Chen et al. [32] introduced an innovative model known as Attention-based Bidirectional Long Short-Term Memory (ABLSTM). This model achieved an accuracy of 97.3% on both a standard public dataset and a dataset collected in a meeting room environment. However, CSI data are typically high-dimensional because they contain amplitude and phase information from multiple subcarriers. This high-dimensional characteristic increases the complexity of the model and computational burden, and it can also lead to overfitting issues. Moreover, the complex relationships within CSI, such as multipath effects, make traditional linear time series models inadequate for effective modeling.

To overcome the aforementioned limitations, researchers have explored methods that combine temporal and spatial approaches. Shang et al. [33] developed a deep learning model that involves integrating WiFi CSI with an LSTM-CNN hybrid architecture for the purpose of recognizing human activities. The proposed framework effectively leverages the temporal sequence properties and spatial features of CSI data by employing LSTM to capture temporal dependencies and CNN to extract local spatial features. This approach addresses the challenges posed by the high-dimensionality, non-stationarity, and complex relationships inherent in CSI data. The effectiveness of this method is demonstrated by an average accuracy of 94.14% on public datasets, highlighting its potential for accurate human activity recognition. Tang et al. [34] proposed a model named Hybrid CNN-GRU (Hybrid Convolutional Neural Network–Gated Recurrent Unit), which significantly improved the performance of human behavior recognition based on WiFi Channel State Information (CSI), achieving an average accuracy of 95.7%. Sheng et al. [35] proposed a deep learning framework that combines CNN and Bidirectional Long Short-Term Memory (BiLSTM), achieving an accuracy of over 95% in cross-scene action recognition tasks. Islam et al. [36] proposed a model named STC-NLSTMNet, which combines depthwise separable convolution and an improved Long Short-Term Memory network (NLSTM) to enhance the performance of human activity recognition. The model achieved an accuracy of over 96% on all datasets.

In recent years, researchers have begun to explore models based on the Transformer architecture, aiming to achieve better performance in various computer vision tasks such as image recognition, object detection, and semantic segmentation. In 2020, Google Research introduced the Vision Transformer (ViT) [37], a novel method that applies the Transformer architecture to image processing. The success of ViT demonstrated that Transformers are not only effective for NLP but can also excel in computer vision tasks. Following this breakthrough, researchers have extended the application of ViT to WiFi-based HAR tasks, leveraging the unique advantages of Transformer models to capture complex spatiotemporal patterns from WiFi CSI. The architecture proposed by Zhou et al. [38] transforms the CSI from Wi-Fi signals into feature points that represent human pose. Subsequently, it employs a self-attention mechanism to effectively parse the spatial information contained within these feature points, accurately reflecting human posture. In this system, the subtle variations in Wi-Fi signals are used to capture human movements, while the self-attention mechanism assists the model in focusing on significant spatial relationships, thereby enhancing the accuracy and reliability of pose estimation.

The research indicates that while some studies have integrated temporal and spatial characteristics, the spatial feature extraction models employed tend to be relatively simple. Therefore, the model we propose improves upon this aspect.

## 3. Dataset

Generally, CSI data consist of a complex vector that include both amplitude and phase information. For our purposes, we use only the amplitude data as input. This is because the raw phase from a single antenna tends to be randomly distributed due to random phase offsets, making the amplitude more stable and more suitable for our recognition tasks. During propagation, high-frequency environmental noise and multipath effects can introduce noise into the raw amplitude of the CSI signal, thereby degrading the final recognition results. Therefore, it is crucial to eliminate this high-frequency noise before feeding the data into the model. Typically, filters or wavelet denoising techniques are employed to process the noise, which constitutes the basic preprocessing of CSI data.

We employed two publicly available datasets: the UT-HAR [25] dataset and the NTU-Fi HAR [37] dataset. Detailed descriptions of these datasets are provided as follows.

### 3.1. UT-HAR

The UT-HAR is the first CSI dataset for HAR. The CSI data were collected by the Intel 5300 CSI tool. [39] UT-HAR utilizes data collected via two Intel 5300 network interface cards, with the transmitting and receiving ends of the device each equipped with three antennas, each antenna pair recording 30 subcarriers. The setup involves a transmission antenna and a receiving antenna placed 3 m apart, with a sampling frequency of 1 kHz. The data collection duration for each action is 20 s, and the dataset includes data from six individuals. After data processing and segmentation by Li et al. [40], the dataset includes seven actions: Lie down, Fall, Walk, Pick up, Run, Sit down, and Stand up. It is an imbalanced dataset, with the detailed counts of all actions listed in the following table (Table 1):

After extracting data from the denoised dataset, we perform normalization to scale the data between 0 and 1. Finally, the processed data are converted into torch. FloatTensor type, resulting in a shape of (len(data), 1, 250, 90), where len(data) represents the number of data entries in the dataset, 1 is the number of channels, 250 denotes 250 data packets in chronological order, and 90 represents the combination of 30 subcarriers across three receiving antennas. The dataset is then divided into different batches, and the input data fed to the model has a shape of (batch, 1, 250, 90).

### 3.2. NTU-Fi HAR

The NTU-Fi HAR is a small dataset, where the CSI data were collected using the Atheros CSI Tool. [41] The setup involved two TP-Link N750 access points (APs), with one functioning as the transmitter and the other as the receiver. Each AP is equipped with three antennas, and each antenna collects data from 114 subcarriers. This dataset encompasses six activities: running, walking, falling down, boxing, circling arms, and cleaning the floor, each activity was recorded for 20 s. For each activity, there are 200 samples, and the dataset includes data from 20 individuals, making it a balanced dataset.

After extracting data from the denoised dataset, we perform normalization on the loaded data, scaling it to a mean of 42.3199 and a standard deviation of 4.9802. Next, we sample the data by taking every fourth data point from a total of 2000 data points, reshaping it into a form of (len(data), 3, 114, 500), where len(data) represents the number of data entries in the dataset, 3 is the number of antennas, 500 denotes 500 data packets in chronological order, and 114 represents the 114 subcarriers for one receiving antenna. The processed data are then converted into torch.FloatTensor type. Finally, the dataset is divided into different batches, and the input data fed to the model has a shape of (batch, 3, 114, 500).

## 4. Method

### 4.1. RGANet

Our model can be broadly divided into two parts: a modified ResNet network serves as the feature extractor and a GRU is used for modeling in the temporal dimension. The combination of residual networks with time series models can lead to significant computational costs, with FLOPs potentially reaching the G level. To mitigate these computational expenses, we have opted to use ResNet18 and GRU as the foundation for our model.

Feature Extractor with Modified ResNet: The ResNet component of our model has been customized to better suit the specific requirements of our task. It not only processes the input data to extract high-level, abstract features that are crucial for understanding the content but also takes into account the spatial characteristics of the input. This means that the ResNet is designed to capture both local and global spatial dependencies within the data. The modifications to the ResNet enhance its ability to train deep networks effectively, leveraging residual connections to mitigate the vanishing gradient problem and ensure that spatial information is preserved and utilized throughout the network.

Temporal Modeling with GRU: After the spatially aware features have been extracted by the ResNet, they are passed on to the GRU. It takes the sequence of spatially enriched features as input and models the dependencies over time, allowing the model to understand the context and relationships between different elements in the sequence. By integrating spatial and temporal information, this part of the model can more accurately predict or classify sequences based on the evolving patterns it detects.

By combining these two components, our model is able to effectively handle complex tasks that require both robust feature extraction, with emphasis on spatial characteristics, and accurate temporal modeling. Our system is as shown in Figure 3.

Table 2 summarizes some of the details of the proposed RGANet model. From the table, it is clear that we have adapted the ResNet architecture to better meet our needs. Our modifications allow the model to extract spatial features while preserving important temporal information. Specifically, we apply convolution and pooling operations only to the spatial dimensions. This enables the model to efficiently capture local spatial features at each time point. This approach is especially beneficial for tasks like understanding human posture or movement patterns. By focusing on spatial features at each time step, the model can recognize and interpret complex movements more accurately. Additionally, we introduced spatial downsampling in the early stages of the network. This reduces the amount of data that later layers need to process, significantly lowering computational costs. Importantly, this design choice ensures that the model retains high temporal resolution, preserving critical time-series information for accurate analysis and prediction.

### 4.2. ResNet_sd

We made modifications based on the standard ResNet18 model.

#### 4.2.1. Modification of Convolutional Kernel Shapes

Convolution kernels (or filters) slide over the input data, performing dot product operations with local regions of the data. This process can automatically identify important patterns or features in the input data without the need for manually designed feature extraction algorithms. For CSI data, convolution kernels can slide over signal matrices to capture changes in signals over time and space. By applying multiple convolution kernels of different sizes and parameters, features at various scales can be extracted. Deep convolutional neural networks are capable of constructing increasingly abstract and complex feature representations. Lower layers typically learn simple features, such as edges or corners; higher layers may learn more complex and discriminative features, such as the overall shape or dynamic changes of specific activity patterns.

After the convolution operation, a non-linear activation function (such as ReLU) is usually applied to increase the model’s expressive power, enabling the network to learn more complex patterns.

Pooling layers reduce the dimensionality of the feature maps. This helps in focusing on the most important features while discarding less relevant details.

Firstly, we modified the shape of the convolution kernels. By using kernels of shape (7, 1) and (11, 1) and applying the convolution operation only along the height dimension, we maintained the completeness of the time series. This approach is essential for capturing the temporal characteristics of actions, ensuring that the temporal information is not lost during feature extraction.

#### 4.2.2. Enhanced Residual Blocks with scSE Module

Secondly, we enhanced the residual blocks by incorporating the scSE (Squeeze-and-Excitation with spatial and channel attention) module [42]. As shown in Figure 4, this module combines spatial attention mechanisms with channel attention mechanisms. The scSE module helps the model focus on important feature regions and channels, thereby improving its representational capacity and performance. This further strengthens the model’s ability to extract spatial features effectively. Importantly, the scSE module operates on the feature maps in a way that does not alter the spatial or temporal dimensions. It applies attention weights to each channel and spatial location, which helps the model emphasize relevant features without disrupting the temporal structure of the data.

Given an input tensor X of shape (B, C, H, W), where B is the batch size, C is the number of channels, and H and W are the height and width, respectively.

The Spatial Squeeze and Excitation (sSE) calculation formula is as follows:(7)$$X_{sse}^{\prime }=X⊙\sigma \left(\mathrm{Conv}2d\left(X;C,1,1\right)\right)$$

For the input tensor X, pass it through a 1 × 1 convolutional layer $\left(\mathrm{Conv}2d\left(X;C,1,1\right)\right)$ to obtain a tensor of shape (B, 1, H, W). Apply the Sigmoid function σ to generate the spatial attention map. Finally, multiply (⊙) the attention map with the original input tensor X element-wise to produce the output tensor $X_{sse}^{\prime }$.

The Channel Squeeze and Excitation (cSE) calculation formula is as follows:(8)$$X_{cse}^{\prime }=X⊙\left(\mathrm{ReLU}\left(\mathrm{Interpolate}\left(Conv2d\left(AvgPool2d\left(X;1\right);C,1,1\right);\left(H,W\right)\right)\right)\right)$$

For the input tensor X, first perform global average pooling across the spatial dimensions using AdaptiveAvgPool2d(*X*;1) to obtain a tensor of shape (*B*, *C*, 1, 1). Then, pass this through a 1 × 1 convolutional layer (Conv2d; *C*, 1, 1) to reduce the channel dimension. Apply the ReLU activation function and interpolate back to the original spatial dimensions (*H*, *W*). Finally, multiply (⊙) the interpolated tensor with the original input tensor *X* element-wise to produce the output tensor $X_{cse}^{\prime }$.

Finally, we process the output results of sSE and cSE by applying each mechanism to the input tensor X. We then combine the outputs by taking the element-wise maximum between the sSE and cSE results, which serves as the final output of the scSE module.(9)$$X^{\prime }=max(X_{cse}^{\prime },X_{sse}^{\prime })$$

#### 4.2.3. Replacement of Standard Convolutions with Depthwise Separable Convolutions

To address the significant computational cost associated with combining ResNet with GRU, we replaced the standard convolutions in the residual blocks with Depthwise Separable Convolutions (DSC) [43]. These consist of a depthwise convolution followed by a pointwise convolution, as shown in Figure 5. Depthwise convolution applies a single convolutional filter to each input channel independently, effectively performing convolution only in the spatial dimensions while keeping the temporal dimension intact. Pointwise convolution: This step uses 1 × 1 convolutional filters to combine the outputs from the depthwise convolution across different channels. While it changes the number of channels, it does not affect the spatial or temporal dimensions. This ensures that the temporal information is preserved. This structure significantly reduces the number of parameters and computational load while maintaining good performance.

In the depthwise convolution, the operation is applied to each input channel independently without mixing information across channels. For an input tensor X with dimensions [H, W, C_in_], where H and W are the height and width and C_in_ is the number of input channels, using a k × k filter size, the output after depthwise convolution will have dimensions [H′, W′, C_in_], with H′ and W′ depending on the kernel size, stride, and padding. The formula for depthwise convolution is as follows:(10)$$O_{i,j,c}=X_{i,j,c}*K_{c}$$
where O is the output; i,j denote the spatial location; c is the channel index; and $K_{c}$ is the convolution kernel corresponding to the cth channel.

Following this, the pointwise convolution is used to mix information across the channels. It uses a 1 × 1 filter to transform the number of channels from C_in_ to C_out_. This step does not change the spatial dimensions of the feature map, so the output dimensions remain [H′, W′, C_out_]. The formula for pointwise convolution is as follows:(11)$$O_{i,j,d}^{\prime }=\sum_{c=1}^{C_{in}}O_{i,j,c}\cdot P_{c,d}$$
where $O^{\prime }$ is the final output, d is the output channel index, and P is the weight matrix that determines how to map from C_in_ input channels to C_out_ output channels.

The computational efficiency comparison between DSC and standard convolution for the same k × k convolution is as follows:(12)$$Efficiency\;Ratio=\frac{Standard\;Convolution\;Cost}{Depthwise\;Separable\;Convolution\;Cost}=\frac{k^{2}\cdot C_{in}\cdot C_{out}}{k^{2}\cdot C_{in}+C_{in}\cdot C_{out}}$$

For large C_in_ and C_out_, this ratio approaches $k^{2}$.

#### 4.2.4. Removal of Fully Connected Layers and Adaptation of Output Shape

Finally, we removed the fully connected layers and modified the output shape to better suit the input requirements of GRU. This change ensures that the spatiotemporal features extracted by the convolutional layers can be seamlessly fed into the recurrent neural network, preserving the temporal dependencies and enhancing the overall efficiency of the model.

### 4.3. Gated Recurrent Unit

Gated Recurrent Units (GRUs) are a type of recurrent neural network (RNN) architecture designed to address the vanishing gradient problem that often occurs in traditional RNNs when dealing with long sequences. GRUs were introduced by Cho et al. in 2014 as a simplified version of Long Short-Term Memory (LSTM) networks [44], which are other popular RNN variants. GRUs aim to capture long-term dependencies in sequential data while being computationally more efficient than LSTM networks.

GRUs simplify the LSTM architecture by merging the cell state and hidden state into a single hidden state and reducing the number of gates from three (input, forget, and output gates in LSTM) to two (reset and update gates). The architecture of the GRU model is shown in Figure 6. Considering the computational capabilities of the device and time efficiency, this paper chose GRU over LSTM as the model architecture.

The internal computation formulas of GRUs are as follows:(13)$$r_{t}=\sigma (W_{r}\cdot \left[h_{t-1},x_{t}\right])$$(14)$$z_{t}=\sigma (W_{z}\cdot \left[h_{t-1},x_{t}\right])$$(15)$$\tilde{h_{t}}=tanh\;(W_{h}\cdot [r_{t}⊙h_{t-1},x_{t}])$$(16)$$h_{t}=\left(1-z_{t}\right)⊙h_{t-1}+z_{t}⊙\tilde{h_{t}}$$
where $r_{t}$ is the reset gate, $z_{t}$ is the update gate, $W_{r}$ is the weight matrix for the reset gate, $W_{z}$ is the weight matrix for the update gate, $h_{t-1}$ is the hidden state from the previous time step, $x_{t}$ is the input at the current time step, σ is the sigmoid activation function, $\tilde{h_{t}}$ is the candidate hidden state, $W_{h}$ is the weight matrix for the candidate hidden state, ⊙ denotes element-wise multiplication, and $h_{t}$ is the final hidden state.

### 4.4. Attention

The attention mechanism is a key component in modern deep learning models, particularly in the fields of natural language processing (NLP), computer vision, and speech recognition. It allows models to focus on specific parts of the input data when making predictions or generating outputs, rather than treating all parts equally. This selective focus helps the model capture long-range dependencies and improve performance on tasks like machine translation, text summarization, and image captioning.

This paper employs a simple additive attention mechanism, similar to the Bahdanau attention mechanism [45], which allows the model to focus on different parts of the input sequence by computing attention weights and using them to create a weighted sum of the input features. This mechanism enhances the model’s performance on tasks that require understanding long-term dependencies or identifying key elements in the input.

The computation formula for this attention mechanism is as follows:(17)$$e_{i}=tanh\;(X_{i}\cdot W+b)$$(18)$$a_{i}=\frac{exp\;(e_{i})}{\sum_{j}exp\;(e_{i})}$$(19)$$output=\sum_{i}a_{i}\cdot X_{i}$$
where $X_{i}$ is the hidden state vector at time step i, W is the weight matrix, b is the bias term, $e_{i}$ is the attention score for time step i, and $a_{i}$ is the attention weight for time step i.

## 5. Experiments

To demonstrate the effectiveness of our proposed RGANet model, we conducted comprehensive experiments on two benchmark datasets: UT-HAR [25] and NTU-Fi HAR [37]. Additionally, we designed a series of ablation studies to systematically evaluate the contributions of each module within the model.

### 5.1. Experimental Parameters and Conditions

The training parameters for our experiment are shown in Table 3. The specifications of the computer used are listed in Table 4.

### 5.2. Comparison with Other Models

To evaluate the performance of the RGANet model, we compare it with several other models that are also based on CSI for HAR. From reference [47], we selected the best-performing spatial feature model, ResNet18, and the best-performing temporal feature model, GRU, as well as a Vision Transformer (ViT) with self-attention mechanisms. Additionally, we chose two models from references [34] and [35] that combine both spatial and temporal feature extraction for comparison. These models are CNN + GRU and CNN + BiLSTM, respectively. This comparative analysis allows us to assess the effectiveness of RGANet against a range of architectures that have been shown to perform well in similar tasks, providing a comprehensive evaluation of its capabilities in the context of CSI-based action recognition.

We adopt well-established performance evaluation metrics, namely accuracy, F1 score, precision, and recall. To address the multi-class classification problem, we use Macro Precision, Macro Recall, and Macro F1 score. The evaluation metrics are outlined as follows:(20)$$Accuracy=\frac{TP+TN}{TP+TN+FN+FP}$$(21)$$Macro\;Precision=\frac{1}{N}\sum_{i=1}^{N}\frac{{TP}_{i}}{{TP}_{i}+{FP}_{i}}$$(22)$$Macro\;Recall=\frac{1}{N}\sum_{i=1}^{N}\frac{{TP}_{i}}{{TP}_{i}+{FN}_{i}}$$(23)$$Macro\;F1\;Score=\frac{2\;*\;Macro\;Precison\;*\;Macro\;Recall}{Macro\;Precision+Macro\;Recall}$$
where *TP* represents the true positive, *TN* the true negative, *FN* the false negative, and *FP* the false positive. N represents the number of classes and i represents the relevant attributes of the ith class of action.

#### 5.2.1. Validation of the UT-HAR Dataset

Table 5 shows the recognition results of RGANet and other models on the UT_HAR dataset.

Based on the provided table, we can conduct a detailed analysis of the performance of each classification algorithm. RGANet is the best-performing model for this task, achieving the highest scores across all evaluation metrics, demonstrating robust classification ability and class balance. ResNet18 and CNN + BiLSTM are also excellent choices, closely following RGANet in performance. They are suitable for applications requiring high precision and high recall. CNN + GRU and ViT, while having slightly lower overall performance, excel in specific metrics (e.g., the recall of CNN + GRU). They can be considered based on specific needs. GRU, although not the top choice for multi-class classification, still provides a solid baseline performance, especially in scenarios where computational resources are limited.

To better illustrate the performance of the RGANet model, we have plotted the confusion matrices for RGANet and the other models as shown in Figure 7.

From the results, it is evident that all models perform relatively poorly in distinguishing between “stand up” and “sit down” actions. This is primarily because these two actions have very similar motion trajectories, differing only in the direction of movement, which makes them prone to misclassification. Additionally, “sit down” is often misclassified as “fall” likely due to the similarity in initial motion patterns.

ResNet18, which relies heavily on spatial features, struggles with actions that involve complex temporal dynamics, such as “stand up” and “sit down”. While this model achieves over 95% accuracy on other actions, it falls short on these two, highlighting the need for temporal modeling.

While the standalone GRU model also performs poorly in classifying “stand up” and “sit down” actions, it misclassifies sitting down as standing up less frequently compared to ResNet18. That said, it is more prone to incorrectly identify sitting down as either falling or lying down. And as we can see, GRU excels in classifying long-term actions like “walk”, but its performance is average for “run”, where it frequently misclassifies “run” as “walk”. This suggests that GRU, while effective at capturing temporal sequences, may not be sensitive enough to distinguish between actions with different speeds or stride lengths. ViT performs better overall compared to GRU, except in the “walk” action.

When comparing CNN+GRU, CNN+BiLSTM, and our proposed RGANet, it becomes clear that RGANet outperforms the other models across all actions. This is particularly true for “sit down”, where RGANet demonstrates superior accuracy. This success underscores the effectiveness of our model in combining both spatial and temporal feature extraction. It can be observed that, despite similarly integrating spatial and temporal feature extraction, our RGANet model comprehensively outperforms the other two models. This indicates that the use of a simple CNN is not adequate for extracting spatial features, thereby validating the effectiveness of our proposed model.

From Figure 8, it can be observed that the model we proposed converges at the fastest rate. ResNet18 and CNN+BiLSTM follow, both stabilizing within 50 epochs. VIT and GRU then stabilize at approximately 150 epochs. Lastly, CNN+GRU exhibits a notably higher training loss compared to the other models, and even with attempts to optimize it by increasing model complexity and adjusting the loss function, its loss value does not reach the levels of the other models. Overall, our model demonstrates superior performance and convergence speed, highlighting its advantage in handling HAR tasks based on CSI.

#### 5.2.2. Validation of the NTU-Fi HAR Dataset

Table 6 shows the recognition results of RGANet and other models on the NTU-Fi HAR dataset.

We can observe that our RGANet model continues to perform exceptionally well. Interestingly, ResNet18 and ViT, which previously showed good performance on other datasets, do not fare as well on this dataset. Conversely, the GRU, which had previously demonstrated lower performance, performs quite well on this dataset. The three models that integrate both spatial and temporal features—CNN + GRU, CNN + BiLSTM, and RGANet—consistently show strong performance across both datasets. The results from both datasets, to some extent, that models which integrate both temporal and spatial feature extraction are effective in the field of CSI data recognition.

To verify the stability of the proposed model, we conducted a 5-fold cross-validation by dividing the dataset into five subsets. The results of this evaluation are presented in Table 7.

The models performed very well on both datasets, especially on the UT-HAR dataset, which demonstrated extremely high stability and accuracy. In contrast, while the NTU-Fi HAR dataset’s average performance was slightly lower than that of UT-HAR, it still reached a high standard and achieved nearly perfect classification results in some folds. However, NTU-Fi HAR exhibited greater performance variability. In summary, RGANet demonstrates outstanding performance across all evaluation metrics, indicating high accuracy and excellent classification performance. It also exhibits a certain level of stability and adaptability to different data distributions. These results suggest that RGANet is likely to be a highly effective classification algorithm suitable for applications requiring precision and reliability. However, it is important to note that these results were obtained under specific datasets and conditions. Therefore, when applying it to other data or scenarios, further validation of its performance may be necessary.

### 5.3. Ablation Study

To demonstrate the role of each component module in RGANet, we have designed a series of ablation studies to analyze the impact of each part on the model’s performance.

#### 5.3.1. The Modified Sections of the ResNet_sd Module

Our modifications to the ResNet_sd module primarily involve adding scSE block within the residual blocks and replacing the convolutional kernels with DSC. To demonstrate the impact of these changes, we use the standard ResNet18 as a baseline for comparison on the UT-HAR dataset.

To validate the effect of the scSE block, we integrated it into the ResNet18, GRU, and VIT models. Figure 9 presents a comparative analysis of the training performance between the original models and their versions augmented with the scSE block.

From Figure 9, it can be observed that the models with the added scSE block converge faster than their original counterparts. This demonstrates that the scSE block enhances the learning efficiency of the models, likely by improving the flow of information and emphasizing important feature channels during training. And, as shown in Table 8, the final recognition accuracy of the models with the added scSE block is also improved; since ResNet18 already has a high recognition accuracy, the scSE module slightly improved the convergence speed of ResNet18 during training. However, this did not result in an increase in the final test accuracy.

We also compared the impact of DSC on ResNet18. The results are presented in Table 9.

From the above, we can observe that by replacing the convolutional kernels with DSC, we significantly reduced the computational cost while simultaneously improving the model’s recognition accuracy. Furthermore, using DSC and scSE block together can further enhance the accuracy, although this comes with a slight increase in computational cost.

The aforementioned results demonstrate that our modifications to ResNet are meaningful. Building on these findings, we will use the modified ResNet network as a feature extraction module and integrate it with GRU and attention mechanisms to form our proposed RGANet.

#### 5.3.2. Impact of RGANet Components and Additional Information About RGANet

We conducted ablation studies on RGANet based on the GRU model, and Table 10 lists the results of various modifications.

The ablation study confirms that the modified ResNet_sd module is the primary driver of performance gains, while the attention mechanism provides additional benefits when integrated into a well-optimized architecture. The full RGANet model, combining all components, achieves the best overall performance, demonstrating the effectiveness of this multi-component approach.

We also conducted a comparative analysis of the computational efficiency of the RGANet model relative to other models, and Table 11 lists the results.

We can see that GRU is the most efficient model in terms of both training and inference time, making it ideal for real-time applications where speed is critical. However, its performance may be limited compared to more complex models like RGANet. The results clearly show that combining GRU with ResNet indeed leads to a significant increase in computational cost, the computational efficiency of our proposed model lies between that of CNN + GRU and CNN + BiLSTM. Certainly, the recognition time for a single sample using our model meets real-time requirements.

Figure 10 illustrates the performance differences among various models, particularly in terms of the balance between accuracy and model complexity. It can be seen that our model achieves the highest accuracy with a relatively small number of parameters, making it well-suited for use on devices with limited computational resources.

## 6. Conclusions

We propose a deep learning model, RGANet, for HAR using CSI signals. The model is composed of two main components: a spatial feature extraction module based on the modified ResNet and a temporal feature extraction module based on GRU. This architecture enables RGANet to effectively capture both spatial and temporal features of CSI signals, significantly enhancing its HAR capabilities. Our model achieves an accuracy of 99.4% on the UT_HAR dataset and 99.24% on the NTU-FI HAR dataset. Compared to other existing models, RGANet shows improvements of 1.21% on the UT_HAR dataset and 0.38% on the NTU-FI HAR dataset. Additionally, we conducted ablation studies to explore the impact of each module within the model on its overall performance.

Furthermore, as mentioned in [48,49], it has been noted that some existing CSI datasets suffer from data leakage issues, where the collected CSI information may inadvertently contain characteristics of individuals. While the focus of this paper is on enhancing the performance of HAR by better integrating spatial and temporal feature extraction, the reliability of the dataset is equally crucial. A reliable dataset can substantiate the model’s generalization ability and the robustness of the results. Therefore, collecting a compliant dataset using WiFi signals is left for future work.

## Figures and Tables

**Figure 1 sensors-25-00918-f001:**
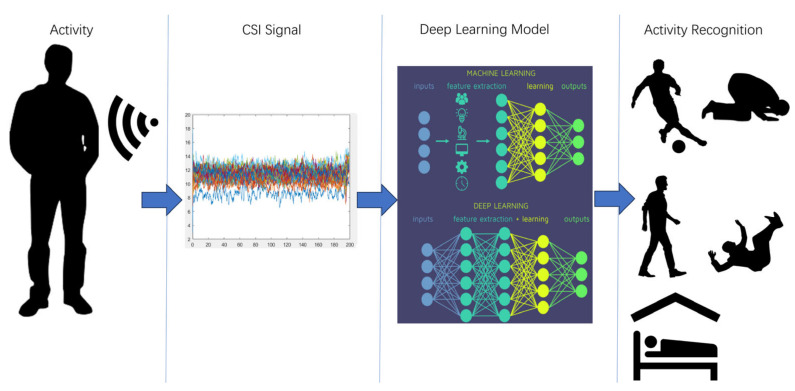
Overview of the CSI-based human activity recognition process.

**Figure 2 sensors-25-00918-f002:**
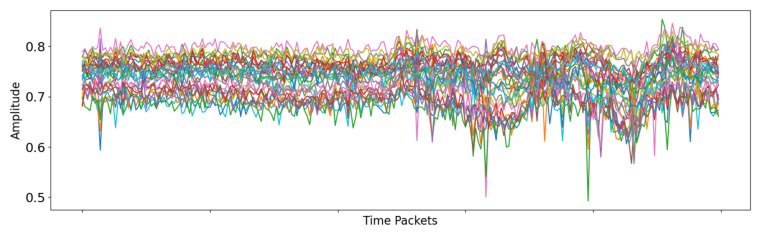
Denoised and normalized CSI carrier waveform for a single-antenna system.

**Figure 3 sensors-25-00918-f003:**
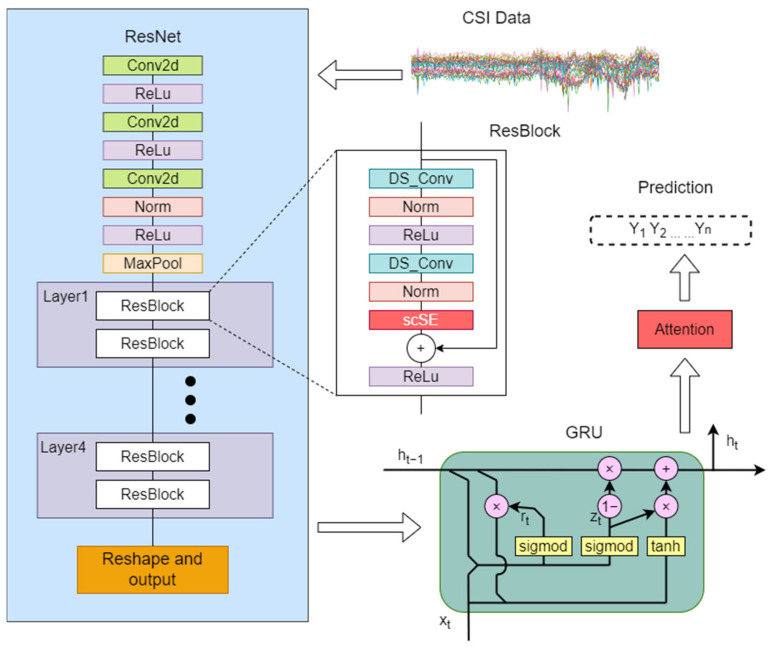
The system framework of RGANet, primarily comprising ResNet and GRU.

**Figure 4 sensors-25-00918-f004:**
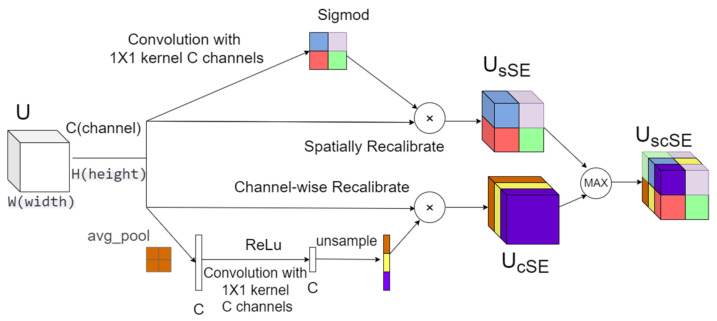
Structure of the Squeeze-and-Excitation with spatial and channel attention model.

**Figure 5 sensors-25-00918-f005:**
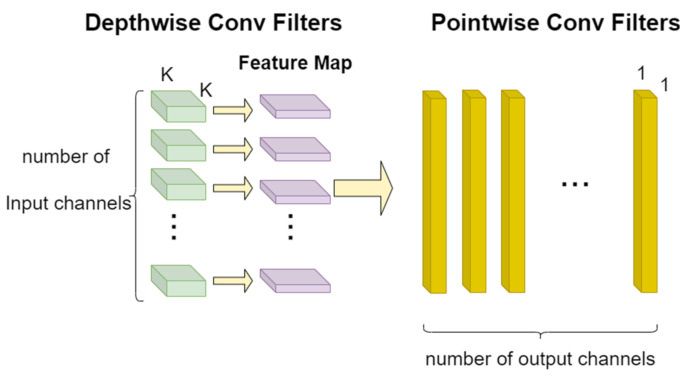
Convolutional filter design of depthwise separable convolutions.

**Figure 6 sensors-25-00918-f006:**
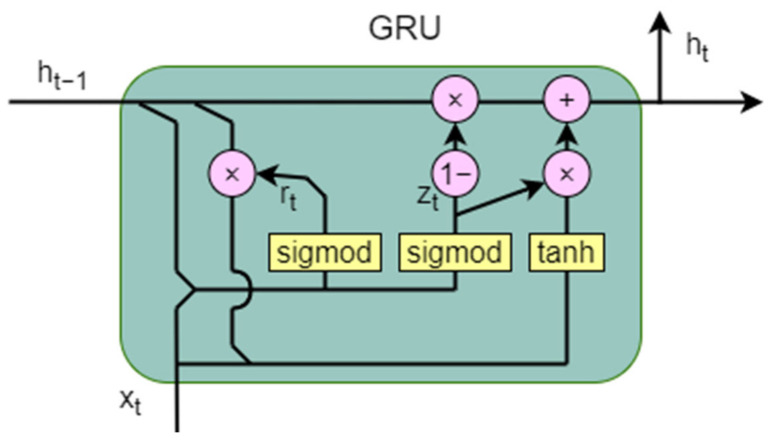
The architecture of the Gated Recurrent Unit.

**Figure 7 sensors-25-00918-f007:**
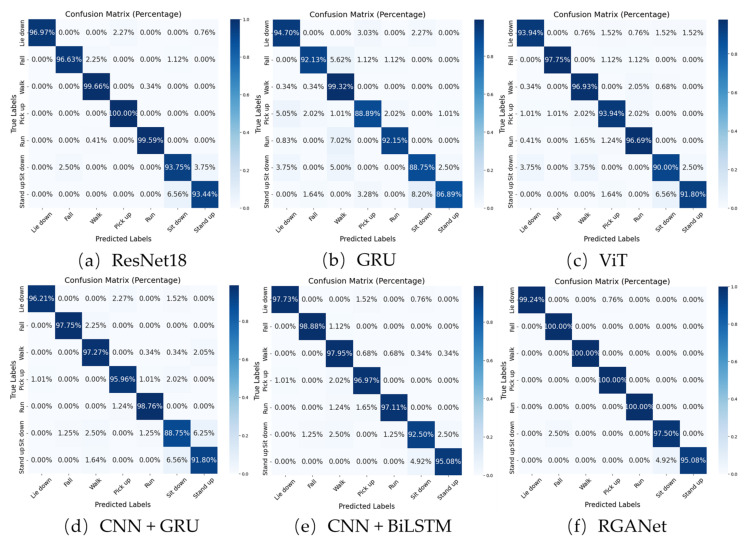
Confusion matrix for six models on the UT-HAR.

**Figure 8 sensors-25-00918-f008:**
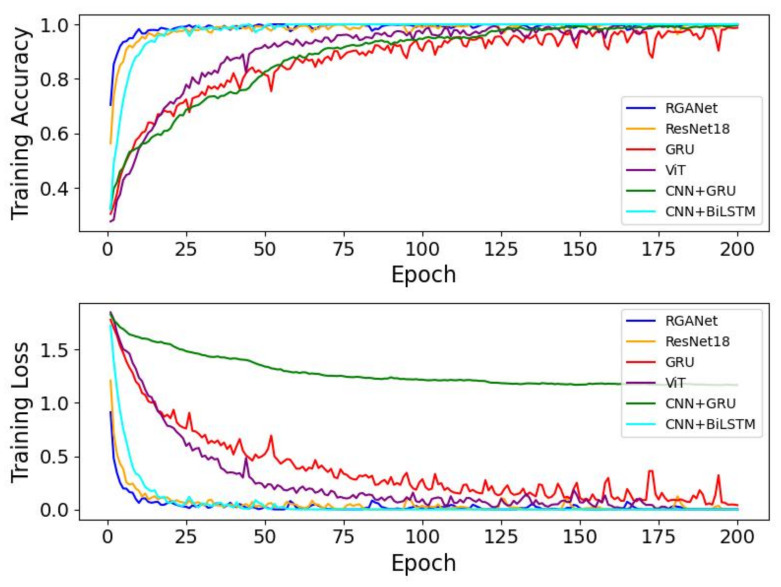
Training curves for RGANet and other models on UT-HAR.

**Figure 9 sensors-25-00918-f009:**
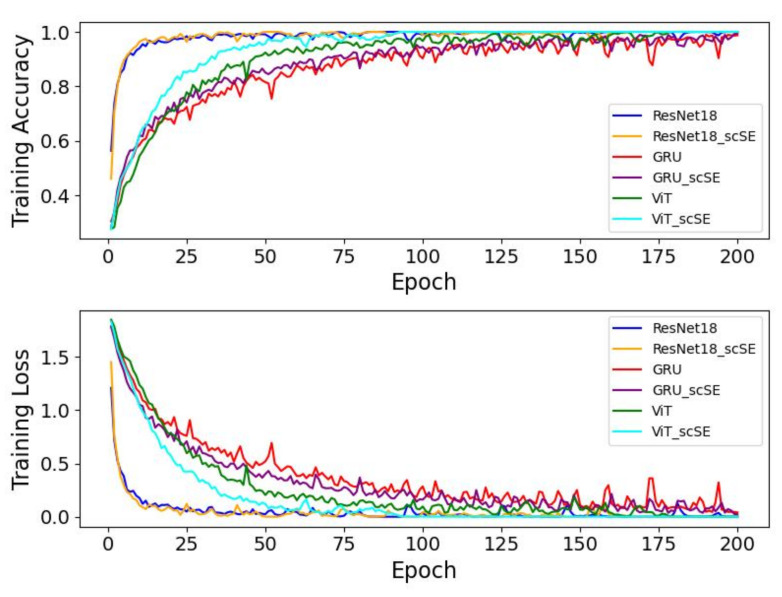
Training curves for models with and without the scSE module on the UT-HAR dataset.

**Figure 10 sensors-25-00918-f010:**
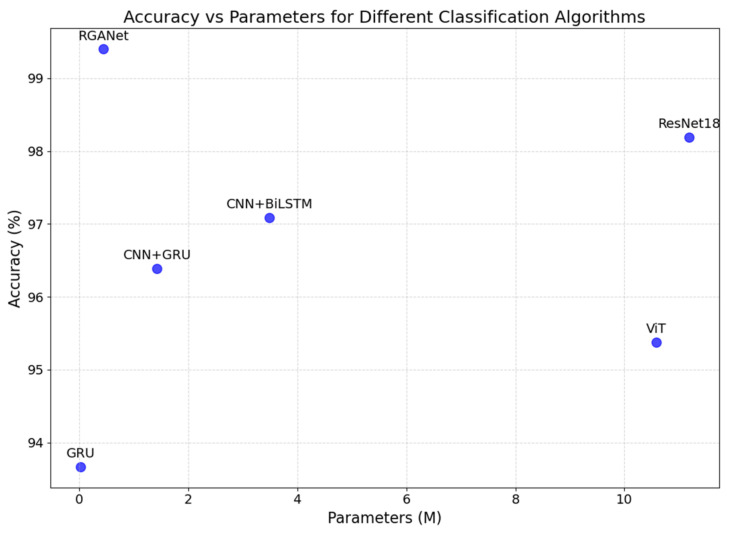
Accuracy vs. parameters for six models on UT-HAR.

**Table 1 sensors-25-00918-t001:** Summary of the UT-HAR dataset.

Activity Name	Number of Samples	Activity Name	Number of Samples
Lie down	655	Run	1121
Fall	442	Sit down	400
Walk	1461	Stand up	304
Pick up	495		

**Table 2 sensors-25-00918-t002:** Summary of RGANet on the UT-HAR dataset.

Section	Layer Type	Output Shape	Parameters
Spatial feature extraction	Conv2d	(batch, 3, 28, 250)	24
	ReLU	(batch, 3, 28, 250)	-
	Conv2d	(batch, 32, 28, 250)	1088
	ReLU	(batch, 32, 28, 250)	-
	Conv2d	(batch, 32, 9, 250)	7168
	MaxPool2d	(batch, 32, 5, 250)	-
	Layer1	(batch, 32, 5, 250)	2816 + 2816
	Layer2	(batch, 64, 3, 250)	9632 + 9728
	Layer3	(batch, 128, 2, 250)	35,648 + 35,840
	Layer4	(batch, 256, 1, 250)	136,832 + 137,216
	Reshape	(batch, 250, 256)	-
Temporal feature extraction	GRU	(batch, 250, 64), (batch, 1, 64)	61,824
	Attention	(batch, 64)	65
Recognition	FC	(batch, 7)	455

**Table 3 sensors-25-00918-t003:** Parameter settings.

Parameter	Value	
Loss function	Cross-entropy	
Optimizer	Adam [46] (β1 = 0.9, β2 = 0.999, epsilon = 1 × 10^−8^)	
Learning rate	0.01	
Batch size	64	
Epochs	UT-HAR: 200	NTU-Fi HAR: 30

**Table 4 sensors-25-00918-t004:** Computer configurations.

Computer System	Windows
CPU	Intel i5-12500H CPU
Memory	16 GB
GPU	NVIDIA GeForce GTX 3060 laptop
Python version	3.9.18
Torch version	2.0.1

**Table 5 sensors-25-00918-t005:** Classification results of different classification algorithms on UT-HAR.

Classification Algorithm	Macro Precision (%)	Macro Recall (%)	Macro F1 Score (%)	Accuracy (%)
ResNet18	97.22	97.15	97.18	98.19
GRU	93.51	91.83	92.60	93.67
ViT	94.73	94.44	94.58	95.38
CNN + GRU	94.50	95.22	94.83	96.39
CNN + BiLSTM	96.46	96.60	96.52	97.09
RGANet	99.01	98.93	98.91	99.40

**Table 6 sensors-25-00918-t006:** Classification results of different classification algorithms on NTU-Fi HAR.

Classification Algorithm	Macro Precision (%)	Macro Recall (%)	Macro F1 Score (%)	Accuracy (%)
ResNet18	96.41	96.21	96.20	96.21
GRU	98.61	98.48	98.48	98.48
ViT	90.62	89.39	89.36	89.39
CNN + GRU	97.02	96.97	96.94	96.97
CNN + BiLSTM	98.90	98.86	98.86	98.86
RGANet	99.28	99.24	99.24	99.24

**Table 7 sensors-25-00918-t007:** Classification results of RGANet on UT-HAR and NTU-Fi HAR datasets.

Dataset	Metrics	Fold					Average
		1st	2nd	3rd	4th	5th	
UT-HAR	Accuracy (%)	99.10	98.99	98.29	99.19	98.89	98.89
	Macro Precision (%)	98.54	98.30	97.76	98.63	98.26	98.30
	Macro Recall (%)	98.26	98.22	97.44	98.44	98.11	98.09
	Macro F1 Score (%)	98.37	98.25	97.60	98.53	98.18	98.19
NTU-Fi HAR	Accuracy (%)	96.21	97.35	98.48	99.62	98.11	97.95
	Macro Precision (%)	96.61	97.57	98.61	99.63	98.30	98.14
	Macro Recall (%)	96.21	97.35	98.48	99.62	98.11	97.95
	Macro F1 Score (%)	96.20	97.34	98.48	99.62	98.10	97.95

**Table 8 sensors-25-00918-t008:** Comparison results of the impact of the scSE block on UT-HAR.

Metrics	Classification Algorithm					
	ResNet18	ResNet18 + scSE	GRU	GRU + scSE	ViT	ViT + scSE
Accuracy (%)	98.19	98.19	93.67	93.78	95.38	95.58

**Table 9 sensors-25-00918-t009:** Comparison results of the impact of DSC and scSE block on ResNet18.

Classification Algorithm	Accuracy (%)	FLOps (M)	Parameters (M)
ResNet18	98.19	49.905	11.182
ResNet18 + scSE	98.19	49.940	11.186
ResNet18 + DSC	98.39	9.907	1.448
ResNet18 + scSE + DSC	98.59	9.131	1.452

**Table 10 sensors-25-00918-t010:** Experimental results of ablation of attention and ResNet_sd modules.

Model Structure	Attention	ResNet_sd	Accuracy (%)
GRU	-	-	93.67
GRU + Attention	√	-	93.88
ResNet_sd + GRU	-	√	99.10
RGANet	√	√	99.40

‘*√*’ indicates that it has a modified network layer and ‘-’ indicates that it does not contain this network layer.

**Table 11 sensors-25-00918-t011:** Computational efficiency of different classification algorithms on UT-HAR.

Classification Algorithm	FLOps (M)	Parameters (M)	Training Time (s) of All Training Samples	Testing Time (ms) of Each Testing Sample
ResNet18	49.91	11.18	5172.13	1.18
GRU	7.60	0.03	1916.92	0.58
ViT	273.10	10.58	21,342.63	9.8
CNN + GRU	39.97	1.43	8591.44	2.86
CNN + BiLSTM	639.48	3.49	27,870.95	10.54
RGANet	164.61	0.44	18,890.93	7.60

## Data Availability

Two datasets were used in this study. The UT-HAR dataset and NTU-Fi_HAR can be found at https://github.com/ermongroup/Wifi_Activity_Recognition, accessed on 7 January 2025; Both datasets are available at https://data.mendeley.com/datasets/dzvgyxkx2f/1, accessed on 7 January 2025. Our trained model and code can be found at https://github.com/h334658994/RGANet, accessed on 7 January 2025.

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
