# Peer review of "RGANet: A Human Activity Recognition Model for Extracting Temporal and Spatial Features from WiFi Channel State Information"

_sensors, 2025, doi:10.3390/s25030918_

Round 1

Reviewer 1 Report

Comments and Suggestions for Authors

This paper presents RGANet, a novel human activity recognition model that leverages CSI signals for feature extraction and activity classification using deep learning techniques. The model comprises a modified ResNet-based feature extraction module and a GRU-based temporal modeling module. By effectively integrating the spatiotemporal features of CSI signals and optimizing the architecture with depthwise separable convolutions and the scSE module, RGANet achieves superior performance on the UT_HAR and NTU-FI HAR datasets while maintaining computational efficiency.

However, a few issues need to be addressed before this manuscript could be accepted.

Major comments:

1. In Section 3.Dataset, the description of the CSI data structure is insufficient. The paper should provide a detailed explanation of the CSI data dimensions, preprocessing methods, and the approach used to construct the data structure. This additional information will help readers gain a clearer understanding of the dataset and its role in the functioning of the proposed model.

2. In Section 4.1.RGANet, the paper does not clearly specify the data structure into which CSI data is constructed as input for the network. This information is critical for understanding how the model processes the data. Additionally, the theoretical rationale for employing convolutional networks to extract features from CSI time-series signals should be further elaborated. This could include a discussion of how convolutional layers are suited for capturing spatial-temporal correlations inherent in CSI signals.

3. To better assess the generalizability of the proposed RGANet model, please consider validating its performance using action data collected from sources other than the UT-HAR dataset. Alternatively, cross-validation with the NTU-Fi HAR dataset could provide valuable insights into the model's robustness and its ability to generalize across different datasets.

Minor comments:

1. The sentences “Nevertheless, interpreting Wi-Fi signals and extracting meaningful insights from them remains a significant challenge. Our research on human activity recognition based on Wi-Fi is indeed one of the applications within the realm of Wi-Fi sensing” (Line 62) and “To more accurately capture the characteristics of the wireless channel, CSI has emerged.” (Line 76) exhibit a conversational tone that may not align with the formal style expected in academic writing. Please thoroughly review this type of issue throughout the article.

2. The arrow at the top of Figure 1 (Line 118) appears visually unbalanced. Consider vertically centering it to achieve improved alignment and a more cohesive visual presentation within the flowchart.

3. Line 214 “(LSTM-CNN architecture” seems to be missing a bracket, please complete it.

4. The variable “i” (Line 444) in Equation (20) is not explicitly explained in the text. Please provide a clear definition of its specific meaning and role within the equation.

Author Response

Comments 1: In Section 3.Dataset, the description of the CSI data structure is insufficient. The paper should provide a detailed explanation of the CSI data dimensions, preprocessing methods, and the approach used to construct the data structure. This additional information will help readers gain a clearer understanding of the dataset and its role in the functioning of the proposed model.

Response 1: Thank you for pointing this out. We agree with this comment. Therefore, we have made additional modifications as suggested, specifically in the section marked in yellow between lines 275-322 of the document.

Comments 2: In Section 4.1.RGANet, the paper does not clearly specify the data structure into which CSI data is constructed as input for the network. This information is critical for understanding how the model processes the data. Additionally, the theoretical rationale for employing convolutional networks to extract features from CSI time-series signals should be further elaborated. This could include a discussion of how convolutional layers are suited for capturing spatial-temporal correlations inherent in CSI signals.

Response 2: Agree, we have added a detailed description of the model's input structure in the previous section that was under discussion. Additionally, we have added the theoretical principles of convolution kernel feature extraction between lines 365-379.

In this section, we detail the underlying theory and mechanisms of how convolution kernels are used to extract features

Comments 3: To better assess the generalizability of the proposed RGANet model, please consider validating its performance using action data collected from sources other than the UT-HAR dataset. Alternatively, cross-validation with the NTU-Fi HAR dataset could provide valuable insights into the model's robustness and its ability to generalize across different datasets.

Response 3: Agree, We have included the results of the 5-fold cross-validation for the NTU-Fi HAR dataset in Table 7. To further elaborate on the implications of our findings, we have added a discussion at line 584.

Comments 4: The sentences “Nevertheless, interpreting Wi-Fi signals and extracting meaningful insights from them remains a significant challenge. Our research on human activity recognition based on Wi-Fi is indeed one of the applications within the realm of Wi-Fi sensing” (Line 62) and “To more accurately capture the characteristics of the wireless channel, CSI has emerged.” (Line 76) exhibit a conversational tone that may not align with the formal style expected in academic writing. Please thoroughly review this type of issue throughout the article.

Response 4: Agree, We have modified it and review the artical, you can see the change at Line 62 and Line 76

Comments 5: The arrow at the top of Figure 1 (Line 118) appears visually unbalanced. Consider vertically centering it to achieve improved alignment and a more cohesive visual presentation within the flowchart.

Response 5: Agree, We have modified it.you can see the change at Line 123

Comments 6: Line 214 “(LSTM-CNN architecture” seems to be missing a bracket, please complete it.

Response 6: Agree, We have modified it.you can see the change at Line 238

Comments 7: The variable “i” (Line 444) in Equation (20) is not explicitly explained in the text. Please provide a clear definition of its specific meaning and role within the equation.

Response 7: Agree, We have modified it.you can see the change at Line 513

Reviewer 2 Report

Comments and Suggestions for Authors

In general, the manuscript is well written. The research topic fits well to the scope of Sensors journal. CSI based human action recognition is a hot and relevant topic in the literature. 

I think the introduction section gives enough background information on the research topic. In my opinion, authors cite major and relevant papers in the related work section. I would have also given background information on RSSI. It is very good that the authors declare the contributions at the end of the introduction section. I have a question to the first contribution. For me, it was not clear what "temporal" and "spatial" features mean in the context of CSI based HAR. The authors published not too long ago a similar paper entitled "VBCNet: A Hybird Network for Human Activity Recognition". It would be good to clarify the difference between the paper and the submitted manuscript. Academic ethics is also important.

I think the description of the proposed method is suitable for a scientific publication. The authors utilized already published elements but they combined them in a novel way. The authors should highlight the motivation behind the chosen elements. I understand that separable convolutions may decrease the computational complexity. But instead of GRU, an LSTM could be also chosen. In Table 2, the number of trainable parameters could be also given. This way, the reader could get a feeling on the computational complexity. It was unclear me which ResNet was utilized. In the literature, I found ResNet-18, ResNet-50, and ResNet-101. I think the experimental results section is mainly OK. The authors compare the proposed method to the state-of-the-art on benchmark databases. On the other, the difference - Table 5 - compared to other methods is not so big. The authors could implement the evaluation methodology published in Exposing Data Leakage in Wi-Fi CSI-Based Human Action Recognition: A Critical Analysis. If one person is completely left out from the training set and allocated to the test set, then the difference between the algorithms is bigger and the overall performance is not so high. Another way could be an accuracy versus number of parameters plot. An analysis on the computational complexity compared with other methods would be also welcomed in the paper.

Author Response

Comments 1: In my opinion, authors cite major and relevant papers in the related work section. I would have also given background information on RSSI.

Response 1: Thank you for pointing this out. We agree with this comment. Therefore, we have made additional modifications as suggested, specifically in the section marked in cyan between lines 125-144 of the document.

Comments 2: I have a question to the first contribution. For me, it was not clear what "temporal" and "spatial" features mean in the context of CSI based HAR.

Response 2: Agree, Temporal Features: Reflect how CSI signals change over time due to human activities. Useful for detecting activities that have characteristic time patterns.

Spatial Features: Reflect how CSI signals differ across space (i.e., different antennas). Useful for determining the location and movement patterns of individuals. we have added it at line 92.

Comments 3: The authors published not too long ago a similar paper entitled "VBCNet: A Hybird Network for Human Activity Recognition". It would be good to clarify the difference between the paper and the submitted manuscript. Academic ethics is also important.

Response 3: To address your concern, we would like to clarify the differences between the current manuscript and the previously published paper entitled "VBCNet: A Hybrid Network for Human Activity Recognition.". "VBCNet: A Hybrid Network for Human Activity Recognition" and the current manuscript were proposed and written by different graduate students. The two papers employ different research methodologies: The paper "VBCNet: A Hybird Network for Human Activity Recognition" is based on research utilizing transformers. In contrast, the manuscript you mentioned is grounded in research employing residual networks (ResNet).

Comments 4: The authors should highlight the motivation behind the chosen elements. I understand that separable convolutions may decrease the computational complexity. But instead of GRU, an LSTM could be also chosen.

Response 4: Agree, The combination of residual networks with time series models can lead to significant computational costs, with FLOPs potentially reaching the G level. To mitigate these computational expenses, we have opted to use ResNet18 and GRU as the foundation for our model.We added it at Line 326

Comments 5: In Table 2, the number of trainable parameters could be also given. This way, the reader could get a feeling on the computational complexity.

Response 5: Agree, We have modified it.you can see the change at table 2( Line 349)

Comments 6: It was unclear me which ResNet was utilized. In the literature, I found ResNet-18, ResNet-50, and ResNet-101.

Response 6: We use ResNet-18, You can see it at line 363

Comments 7: On the other, the difference - Table 5 - compared to other methods is not so big. The authors could implement the evaluation methodology published in Exposing Data Leakage in Wi-Fi CSI-Based Human Action Recognition: A Critical Analysis. If one person is completely left out from the training set and allocated to the test set, then the difference between the algorithms is bigger and the overall performance is not so high. Another way could be an accuracy versus number of parameters plot. An analysis on the computational complexity compared with other methods would be also welcomed in the paper.

Response 7: we agree with this comment. But the dataset used in our study is sourced from public datasets, which does not provide distinct annotations for individual participants, so we can’t implement the evaluation methodology published in Exposing Data Leakage in Wi-Fi CSI-Based Human Action Recognition. Additionally, we have added a figure showing the relationship between accuracy and the number of parameters at line 655.

Round 2

Reviewer 2 Report

Comments and Suggestions for Authors

I think the manuscript can be accepted now. The authors answered my questions and concerns. Figure captions could be longer since they are not too informative. For instance, the captions of Figures 1 and 2 are very short. Capitals are also not used consistently in captions (Figure 2).

Author Response

Comment 1:Figure captions could be longer since they are not too informative. For instance, the captions of Figures 1 and 2 are very short. Capitals are also not used consistently in captions (Figure 2).

Response 1:Thank you for pointing this out. We agree with this comment. Therefore, we have made additional modifications as suggested, you can find it at line124,154 348,396,427,620,655